# Criteria and Protocol: Assessing Generative AI Efficacy in Perceiving EULAR 2019 Lupus Classification

**DOI:** 10.3390/diagnostics15182409

**Published:** 2025-09-22

**Authors:** Gerald H. Lushington, Sandeep Nair, Eldon R. Jupe, Bernard Rubin, Mohan Purushothaman

**Affiliations:** Progentec Diagnostics Inc., Oklahoma City, OK 73104, USA; sandeepnair@progentec.com (S.N.); ejupe@progentec.com (E.R.J.); brubin@progentec.com (B.R.); mpurushothaman@progentec.com (M.P.)

**Keywords:** American College of Rheumatology (ACR), antinuclear antibody (ANA), disease classification, European Alliance of Associations for Rheumatology (EULAR), generative artificial intelligence (genAI), systemic lupus erythematosus (SLE), natural language processing (NLP), undifferentiated connective tissue disorder (UCTD)

## Abstract

**Background/Objectives:** In clinical informatics, the term ‘information overload’ is increasingly used to describe the operational impediments of excessive documentation. While electronic health records (EHRs) are growing in abundance, many medical records (MRs) remain in legacy formats that impede efficient, systematic processing, contributing to the extenuating challenges of care fragmentation. Thus, there is a growing interest in using generative AI (genAI) for automated MR summarization and characterization. **Methods:** MRs for a set of 78 individuals were digitized. Some were known systemic lupus erythematosus (SLE) cases, while others were under evaluation for possible SLE classification. A two-pass genAI assessment strategy was implemented using the Claude 3.5 large language model (LLM) to mine MRs for information relevant to classifying SLE vs. undifferentiated connective tissue disorder (UCTD) vs. neither via the 22-criteria EULAR 2019 model. **Results:** Compared to clinical determination, the antinuclear antibody (ANA) criterion (whose results are crucial for classifying SLE-negative cases) exhibited favorable sensitivity 0.78 ± 0.09 (95% confidence interval) and a positive predictive value 0.85 ± 0.08 but a marginal performance for specificity 0.60 ± 0.11 and uncertain predictivity for the negative predictive value 0.48 ± 0.11. Averaged over the remaining 21 criteria, these four performance metrics were 0.69 ± 0.11, 0.87 ± 0.04, 0.54 ± 0.10, and 0.93 ± 0.03. **Conclusions:** ANA performance statistics imply that genAI yields confident assessments of SLE negativity (per high sensitivity) but weaker positivity. The remaining genAI criterial determinations support (per specificity) confident assertions of SLE-positivity but tend to misclassify a significant fraction of clinical positives as UCTD.

## 1. Introduction

Healthcare fragmentation can pose major impediments to the effective treatment of chronic illness, especially for patients with lengthy medical histories who have transitioned across different providers or who require the attention of multiple medical services [1]. A crucial barrier is the limited interoperability in medical records, such that providers caring for a given patient are faced with the task of manually ascertaining care-critical information from documents in disparate formats [2].

One medical discipline where fragmentation is especially problematic relates to the care for intractable disorders such as systemic lupus erythematosus (SLE) [3], for which informed treatment is often contingent on harvesting valuable insight from case histories. Manual processing of voluminous, long-duration medical records (MRs) encoded in legacy formats can overlook valuable insight—a problem relating to what is known both informally and formally as ‘information overload’ [4,5,6,7]. Such problems may prove addressable using generative artificial intelligence (genAI) to intelligently automate document parsing [8], but doing so will first require characterizing the incumbent challenges and opportunities. A recent review by Sequí-Sabater and Benavent [9] surveyed the adoption of AI within the rheumatological practice but focused more on machine learning and deep learning, citing genAI as only a support tool for research, communications, and clinical decisions, without mentioning MR deconvolution. This suggests, as Sequí-Sabater and Benavent imply [9], that serious consideration for how genAI may impact medical practice is just beginning.

### 1.1. Prospective Roles for AI in Medical Record Extraction

Hindering the medical adoption of genAI is concern over the risk and reproducibility [8,10,11] relating to the stochastic underlying algorithm., i.e., multiple replicate runs of the same code reading the same documents may yield different results. This limits the range of target applications to support functions which, to make a human analogy, value ‘subjective perception’ as much as ‘objective precision’. Thus, using genAI to enumerate and sort evidence may attain greater trust than automatically ranking observations or proposing conclusions. Further, this conservative approach aligns well with addressing MR-driven information overload. Ideally, genAI document parsers may soon tame the overload problem associated with bulky MRs and do so according to physician-specified goals at reliable levels of specificity, sensitivity, and consistency.

### 1.2. Potential Applications for Rheumatology

GenAI-augmented information extraction may prove broadly applicable across medicine, but there is value in beginning with well-defined protocols based on specific observations, tests, or medication profiles of established value to well-defined medical objectives. As a discipline well-rooted in information-intensive decision-making [12,13], rheumatology abounds in such systematic protocols. Examples include well-validated criterially specific protocols to assess or classify rheumatoid arthritis [14], scleroderma [15], Gout [16], fibromyalgia [17], psoriatic arthritis [18], idiopathic inflammatory myopathies [19], and systemic lupus erythematosus (SLE) [20]. These protocols encode statistically significant cross-correlations among laboratory tests, imaging, qualitative clinical observations, patient feedback, etc., relative to patient status markers for a given disease or treatment. With documented clinical sensitivities and specificities all ranging between 80 and 99% [14,15,16,17,18,19,20], derived from moderate criterial bases (ranging from three to roughly two dozen terms), such protocols offer viable, potentially excellent targets for genAI emulation.

### 1.3. Applications to SLE Classification Criteria

In a recent pilot demonstration [21], we assessed the Claude-3-Haiku large language model (LLM) [22] as a means for mining MRs for criteria associated with the 1997 American College of Rheumatology (ACR) (referred to herein as ‘ACR 1997’) SLE classification protocol [23]. Medical visit annotations and test results were analyzed for 78 patients, including 46 with eventual SLE-positive (SLE+) clinical classifications, 18 consigned to ‘undifferentiated connective tissue disorder’ (UCTD), and 14 cases with inadequate evidence of SLE or adjacent conditions (SLE-).

The pilot study provided proof-of-principle insight into genAI suitability for mining medical criteria, but increased utilization of the 2019 joint ACR/EULAR classification standard [20] (hereafter called EULAR 2019) has motivated this sequel study. ACR 1997 and EULAR 2019 achieve similar levels of predictive specificity (both roughly 93.4%) [20,23], but EULAR 2019 achieves superior sensitivity (96.1%) compared to ACR 1997 (82.8%) despite considering similar core criteria. The difference in sensitivity arises from three significant adjustments to the decision-making structure: EULAR 2019 derives greater sensitivity from finer criterial graining (22 relatively narrow criteria, replacing 11 broader terms), trained criterial weighting, and specification of the antinuclear antibodies test (ANA) as an entry criterion [20]. GenAI determinations should mimic and ideally match such superior clinical performance. To realistically assess a general genAI method for such challenges, our updated study emulates the EULAR 2019 protocol in translating real-world medical histories among candidate SLE patients toward preliminary ANA-related triage and the broader criterial differentiation of SLE+ and UCTD cases.

## 2. Materials and Methods

To establish preliminary performance benchmarks, genAI was employed to assess the MRs of 78 individuals according to evidence mapping to the 22 criteria associated with EULAR 2019 classifications. Specific details follow.

### 2.1. Available Medical Records

Medical records from two prior clinical studies [24,25] were used in a manner consistent with IRB (institutional review board) specifications (corresponding approvals appended). For the set of patients retrospectively examined for this analysis, details such as demographic composition, recruitment, outcomes, etc., are reported elsewhere [21]. For the purposes of context, the current study focused on 78 individuals, divided equally among two distinct groups:A total of 39 individuals whose pre-classification case histories [24] (hereafter called ‘pre-SC’) covered 1+ years, ending with clinical evaluation for SLE, via either the ACR 1997 or EULAR 2019 protocols.A total of 39 individuals with confirmed SLE cases [25] (hereafter called ‘post-SC’) covering 1+ years, all beginning at some unspecified duration after prior SLE classification.

MRs for all cases included visit annotations and medical test results, all provided as scanned text or tabular documents, converted to flat electronic text using the Textract program for optical character recognition on AWS [26]. All details relating to the acquisition, secure handling, and processing of documents follow our protocol published previously [21]. For pre-SC records, documented clinical determinations were provided for some SLE classification criteria, including 15 classifications using EULAR 2019 [20] and 24 via ACR 1997 [23].

### 2.2. EULAR 2019 Criteria

For reference, all classification criteria are listed in Table 1. Among these, clinical determinations (explicit or inferred) were available to validate ANA predictions for all 78 cases; however, only 15 cases contained clinical validation for the remaining criteria.

Notable for EULAR 2019 SLE classification criteria is the data heterogeneity, driven by disparate formats across quantitative test results, descriptive observations, medical imaging, etc. Record standards also vary broadly among practices. ANA determinations, for example, may be conveyed numerically (e.g., ‘ANA titer 1:160’), qualitatively (‘positive ANA result’), or descriptively (‘speckled ANA’), and similar variance exists for other criteria. To address this, genAI can benefit from preparative strategies such as augmented-generation (AG) techniques [27] that supplement prompts with extensive related text examples to steer LLM contextualization.

In our case, lacking a large compendium of sample SLE MRs, the alternative is ‘prompt engineering’ [28], where representative phrasing and reporting formats augment the core extraction query specification. Specifically, the rigors of organic LLM training may be supplemented (or, to an extent, replaced) by biased techniques associated with earlier natural language processing (NLP) methods such as intelligent (i.e., discipline-biased) specification of keyword and keyphrase lists that populate manually dictated genAI prompts. Such keys can be evaluated within the test documents (e.g., medical records to be mined) via rule-based or lexicon-based assessment, thus guiding new iterations of prompts that balance the original discipline-biased vocabulary with terms that are actually representative of the target documents.

### 2.3. Search Parameters and Prompt Specifications

Most genAI text retrieval and assessment protocols in this study are identical to those described in detail in our prior study [21] on ACR 1997 criteria [20], basically involving a two-pass assessment of all criteria, where the first pass condenses all available clinical notes, lab test results, imaging reports, ICD-10 codes, and structured and unstructured data formats down to a body of potentially relevant annotations, upon which the second pass performs rigorous relevance scoring. Variations in criterial performance between the current study and the precursor are attributable to use of a different LLM (Claude 3.5, San Francisco, CA, USA [29] for the current study; Claude 3.0, San Francisco, CA, USA [22] for the prior work [21]) and different criterial definitions determined by the clinical decision protocol encoded for the EULAR 2019 [20] and ACR 1997 [23] studies.

There is fairly strong mapping between the core content of the eleven ACR 1997 criteria and the 22 criteria from EULAR 2019, but many sub-criterial factors in ACR 1997 classification are represented as full criteria in EULAR 2019, thus altering various Boolean constructs in our genAI prompts. These modifications include, higher emphasis on correct determination of patient ANA status has prompted adjustment of the first-pass NLP prompt in the current study to be more permissive. In particular, any mention of ANA titer, pattern, or test, regardless of value, is grounds for first-pass compilation, while strong exclusionary language was discarded from the earlier study such as, for example, restrictions aimed at disqualifying ANA results associated with drug-induced lupus or any other conditions unrelated to SLE.

### 2.4. Criterial Assessment Procedure

Criterial classification roles are shown in a schema (Figure 1) that empowers high sensitivity and specificity for SLE classifications [20]. Relative to the ACR 1997 framework, the EULAR 2019 stipulation of ANA as an entry criterion often reduces clinical effort by short circuiting the exhaustive determination of criteria 2–22 (Table 1) for ANA- cases. Regardless of ANA status, however, physicians may derive value from genAI-based clinical support across the fuller set of analytical determinations. For example, a current SLE− determination is overruled by prior ANA+ evidence, even buried deeply within a patient’s medical history. Also, various EULAR 2019 SLE criteria may shed light on alternative diagnoses, even given ANA− status. Our protocol is thus configured to assess all 22 criteria regardless of ANA determination.

NLP criterial assessments were conducted over five replicate runs, aimed at gauging internal consistency of genAI stability as a function of prompt composition and generative sampling governed by stochastic temperature and token size [22]. Consistency over the five replicates was scored according to the trivial implementation of Johnson Lindenstrauss random projection [30]: full consistency (all five replicate determinations agreed with each other) was scored as 1.0, partial consistency (four replicates agreed but one dissented) was scored as 0.6, and weak consistency (three replicates agreed; two dissented) was scored as 0.2.

For ANA (criterion 1), prediction statistics were computed over all 78 cases and were assessed for known or inferred clinical ANA status. Predicted ANA outcomes fed into the EULAR framework (top of Figure 1) as a basis for SLE- triage. Prediction statistics for all other criteria (those numbered 2–22 in Table 1) were computed for all 27 pre-SC cases with full criterial clinical determinations and were applied toward UCTD vs. SLE+ discrimination, both for categorical branching and final SLE scoring (see Figure 1).

As indicated previously, aspects of the assessment protocol employed in the current study emulate our prior publication [21] and are not repeated verbatim, but a key enhancement is criterial influence analysis, specifically aimed at discriminating UCTD and SLE+ cases according to disproportionate criterial prevalence and weights differentiating the two classes. Analysis was thus conducted to quantify differential weighted influence of each individual criterion relative to all criteria involved in SLE+ determinations (Equation (1a)), and the weighted influence of each individual criterion relative to all criteria involved in UCTD determinations (Equation (1b)), as follows:(1a)$$Influence\left(S\right)\;=\;\frac{N_{s}W_{s}}{\sum_{i\in \text{\{}\mathrm{crit}\left(S\right)\text{\}}}\left(N_{i}W_{i}\right)}$$(1b)$$Influence\left(U\right)=\frac{N_{u}W_{u}}{\sum_{i\in \text{\{}\mathrm{crit}\left(U\right)\text{\}}}\left(N_{i}W_{i}\right)}$$
where ‘*S*’ refers to a given criterion (Table 1 criteria 2–22) with influence on the determination of SLE+ cases, *N_S_* is the number of total counts of that criterion across the set of SLE+ cases for which criteria were clinically determined, *W_S_* is the EULAR-2019 weight of that criterion, ‘*U*’ refers to a given criterion with influence on determining UCTD cases, *N_U_* is the number of total counts of that criterion across the set of UCTD cases for which criteria were clinically determined, and *W_U_* is the corresponding weight of that criterion. Denominator summations include all similarly defined criterial count-weight products for any of the criteria with non-zero counts, either among SLE+ cases (Equation (1a)) or for UCTD cases (Equation (1b)).

### 2.5. Statistical Analysis

For standard errors of criterial assessments, it was noted from Wald interval analysis and Wilson Score interval analysis [31] that, to 95% confidence, no criteria had an approximately even balance between the ratio of positive clinical determinations versus the ratio of negative clinical determinations. This precludes the presumption of binomial distribution (e.g., Wald analysis) in determining standard error. Consequently, 95% confidence intervals for all derivative performance statistics (i.e., accuracy, sensitivity, specificity, positive predictive value, negative predictive value, and weighted criterial influence) were based on Wilson Score interval determination. Considerations of the small sample sizes prompted the use of the Fisher exact test for qualitative assertions of certitude, rank, or superiority, all of which employed a *p*-value of 0.05 as the upper limit for null-hypothesis rejection.

## 3. Results

Broader details of clinical and predictive statistics are provided for all EULAR 2019 criteria in Appendix A, but, for brevity, we focus herein on a select subset chosen for significance relative to our main foci of ANA-based triage and the differentiation of SLE+ and UCTD classes. Table 2 below thus targets eight criteria of prospectively elevated clinical and/or genAI impacts.

Notably, while genAI predictive metrics varied substantially for different criteria, the summary of the statistics for all 22 criteria (bottom row) reveals that the specificity and NPV are greater than the sensitivity and PPV (*p* < 0.02). The diminished sensitivity and PPV scores reflect a low incidence of clinical positives, i.e., only four criteria (ANA, NSA, oral ulcers, and joint involvement) were clinically identified in 10+% of cases.

### 3.1. ANA Results

The ANA summary statistics diverged from other SLE criteria by demonstrating significant rates for sensitivity (95% CI: {0.68,0.87}) and PPV (95% CI: {0.77,0.93}), while specificity and NPV were marginal. ANA divergence largely arose from the greater determination among studied cases and higher clinical positivity. Given its unique weight in SLE classification, raw ANA results are outlined in Table 3 to illustrate the origins of this trend.

Table 3 indicates that high TP rates underlie the strong sensitivity and PPV scores. Conversely, TN, FN, and FP counts are all smaller and have a similar magnitude, thus suppressing the specificity and NPV. These latter deficiencies (see Table 2) were driven by 14 post-SC cases (all clinically confirmed SLE+) for which manual scrutiny of the MR documents revealed that the genAI first pass generally uncovered ANA status indicators; however, the more stringent second pass scoring was less reliable in confirming that these cases legitimately met EULAR 2019 standards, largely due to cases where relevant records of ANA testing that produced positive determinations occurred prior to the one to two years of compiled medical history made available to this study. Conversely, there were five pre-SC cases where computational analysis yielded ANA+ indications with supporting quantitative evidence (1:40 titers) that would have been rejected by human scrutiny, plus three other records stating ANA positivity but lacking quantitative corroboration. The final putative FP appears to be an instance where MR evidence contained a clear indication of ANA positivity at 1:80 titers that, apparently, was overlooked during the clinical SLE classification.

Despite criterial deficiencies, genAI findings and clinical SLE classifications led to similar distributions of predictive outcomes. Figure 2A (plotting case attrition after each decision branch) reveals analogous trends (upper portion of Figure 2A) for both real clinical evaluation (blue) and genAI simulation (brown). Final clinical and genAI classification ratios (Figure 2B) suggest a rough similarity in the ratios of SLE− to UCTD to SLE+ classifications, even if class concordances (colored heatmap) are marginal. In total, genAI achieved an exact match with clinical classifications in 54% of cases (predominantly SLE+ true positives), while the dominant mismatch (18% of all cases) involved clinical SLE+ cases for which genAI predicted SLE−, dominated by the aforementioned ANA false negatives.

### 3.2. Criteria Differentiating UCTD Versus SLE+

In comparing clinical SLE+ vs. UCTD classifications, our analysis reveals some inter-class variation in the relative prevalence of positive criterial determinations. While formal clinical determinations of many criteria are sparse (see Appendix A, Table A1), some criteria (e.g., joint involvement) are well documented in MRs, thus potentially adding insight that, prior to SLE classification, could help to inform clinical priorities. To such ends, Table 4 details the relative influence (see Equations (1a) and (1b)) for selected genAI criterial determinations.

The analytical value of Table 4 may reside in documenting those metrics with influences that differentiate SLE+ from UCTD and vice versa. For this, the differential influence of anti-Smith antibodies (ADS) (*p* < 0.004) and low C3/C4 (*p* < 0.03) statistically favored clinical SLE+ determinations, while proteinuria (*p* < 0.009) and non-scarring alopecia (NSA) (*p* < 0.015) had elevated representation in UCTD determinations.

## 4. Discussion

In healthcare and biomedicine, genAI has emerged as a proverbial solution in search of an application. The incumbent algorithms have been tested extensively relative to tasks such as MCAT test-taking [32] or medical diagnoses [33], achieving a performance that seems intriguing but consistently fails to attain levels required for stringent medical practice [32,33]. While genAI may still rise to parity with medical experts, it may be that other medically important objectives will prove more readily attainable. In particular, this study seeks to assess the potential value of genAI toward alleviating information overload scenarios that exacerbate the widespread problem of fragmented care for chronic disease.

Clinical classification of SLE often presents laborious demands for patient evaluation and lab testing. As outlined in Figure 1, the EULAR 2019 framework presents opportunities to reduce this burden via decision branches, including the triage of SLE cases based on failure to achieve the ANA entry criterion, followed by applying a ‘first-to-ten’ pursuit of SLE+ (i.e., a patient is classified with SLE upon reaching ten criterial points, regardless of the total number of criteria evaluated), provided at least one criterion is ‘clinical’.

As discussed earlier, the advantage of this protocol is that decisions may be reached well before all criteria are fully assessed. A possible disadvantage, however, is that some patients may end up with incorrect SLE− or UCTD assessments if useful criterial evidence from past medical history is overlooked—a risk exacerbated by long MRs in legacy formats. Automated MR mining via genAI may help mitigate this risk.

### 4.1. GenAI Progress Toward ANA Determination

As observed in our Results, genAI tends to accrue promising measures of sensitivity and PPV, thus suggesting an encouraging performance for characterizing the analytically crucial criterion of ANA status. Unfortunately, MR documents posed challenges such as limited duration—a source of numerous ANA false negatives that suppressed NPV performance. This might suggest an imperative to statistically profile the frequency of ANA testing among classified SLE patients to potentially infer a minimum temporal MR duration to mine in order to achieve set levels of predictive confidence.

A second concern arises from the ANA results (see Table 2 and Table 3) whose statistical specificity (0.57; 95% CI: {0.49,0.70}) reflects a substantial number of false positives (9). Notably, this specificity level aligns with clinical assessments for indirect immunofluorescence testing [34,35], although more recent ELISA tests have a substantially better specificity [34,36]. MR annotations might also report ANA+ status without proximal indication that the positivity was actually at the 1:40 titer level, which is often considered to be a marker for possible autoimmune dysfunction but is generally inadequate for SLE classification. This weakness sometimes relates to limits in token-based contextualization (i.e., conceptual modifiers to a given fact may not influence conclusions about that fact if the *fact-modifier* and *modifiable-fact* are not both within the same text window under consideration within granular genAI processing) but may also reflect the residual discussion of an older test result whose precise details (potentially involving a low-precision test or a 1:40 titer) predate the one- to two-year MR window available for genAI analysis in this study.

### 4.2. GenAI Progress Toward Criterial Discrimination Between UCTD and SLE+

An interesting insight emerged from profiling potential roles of non-ANA criteria in discriminating between SLE+ cases and those still best described as UCTD. Specifically, evidence in Table 4 pinpoints metrics that have weighted genAI prevalences, suggesting a differing prevalence in SLE+ cases versus UCTD. Specifically, the computed influence of anti-Smith antibodies (ADS) (*p* < 0.004) and low C3/C4 (*p* < 0.03) discriminated significantly in favor of SLE+ over UCTD, while proteinuria (*p* < 0.009) and non-scarring alopecia (NSA) (*p* < 0.015) had statistically greater weighted genAI prevalences in UCTD than SLE+.

### 4.3. Sampling Limitations

While this preliminary evidence is interesting, it is important to recognize sampling limits. SLE+ sampling is modestly robust (48 patients), but limited SLE- sampling (22 patients) hinders the full evaluation of the capacity of genAI to distinguish SLE from other non-rheumatological maladies, and too few UCTD outcomes (nine cases) are available to assure the representation of the breadth of connective tissue pathological heterogeneity. Furthermore, while we have access to definitive SLE classifications for all 78 cases examined in the study, and we have examined one- to two-year compendia of MRs for all, our access to explicit clinically assessed criterial-level determinations is limited to 15 cases at the EULAR 2019 level, plus 12 cases for which simpler ACR 1997-level criterial assessments were made. This collectively means that, while our total sample is fairly robust for evaluating the overall prospect for genAI recognition of true SLE+ cases, we have inadequate, fine-grained symptomatic points of validation to confidently differentiate UCTD cases versus SLE- or to be confident that genAI truly identifies specific disease measures. Thus, the initial significance of genAI-derived SLE classifications is initially promising but warrants further investigation for clear characterization of UCTD cases and confident prediction of SLE- status. Furthermore, many criterial-level predictions are not statistically significant; thus, firm criterial conclusions will require the assessment of a larger cohort.

## 5. Conclusions

Generative AI methods are gaining attention for diverse medical information tasks [8,37,38,39,40], with demonstrable prowess in technical extraction across many disciplines [41,42,43,44]. Within the medical community, however, uncertainty persists regarding whether genAI has risen to the point of reliable decision support and, if so, what roles could be confidently delegated [37,45,46,47]. Achieving such trust requires accepted benchmarks and extensive validation.

In order to extend our prior pilot assessment of genAI replication of the SLE ACR 1997 classification [21], the present study furthers the discourse by considering genAI efficacy for parsing legacy MRs (i.e., digitized paper documents) as a means for characterizing patients according to their case histories, with performance being assessed based on the propensity of genAI to replicate clinical determinations to support the classification of prospective SLE patients according to the EULAR 2019 protocol.

To this end, net performance statistics for resolving SLE+ versus UCTD versus SLE- determinations achieved a 54% accuracy ±11% (95% CI) for tripartite classification. This is within the performance range determined by meta-analysis on the use of genAI for diagnosing various medical conditions [33,48], but it can be objectively surmised that none of these studies fully merit widespread practical medical adoption yet.

Advancing from numbers that are marginally promising but functionally inadequate may be supported by the continued improvement of genAI algorithms, but SLE case heterogeneity [49,50] suggests that informatics alone will not fully close the gap. Immediate next steps thus involve assessing which SLE-relevant metrics were handled well by the current methodology, determining why these successes occurred (e.g., which MR instances and data structures tended to accurately document specific criteria) and correspondingly adapt our strategies for information acquisition. While we believe that the current version represents a useful incremental benchmark based on straightforward evidence, it is equally apparent that genAI may be powerfully augmented by complementary algorithms operating on data other than what genAI or other NLP methods might consider. A potentially impactful augmentation, for example, could strengthen associative reasoning by also mining MRs for intervention history. Key data may include procedures, tests, and multiple aspects of prescription history (e.g., dose, compliance, apparent effectiveness, etc.). Such data generally resides outside of conventional digital diagnostics but adds a context that, via machine learning or deep learning, may illuminate statistical associations with specific disease states, whether they be previously diagnosed, suggested, or unsuspected. Such associations would not be considered actionable medical evidence but may prove valuable for assisting medical professionals in identifying lines of inquiry and for filling MR gaps resulting from care fragmentation.

While genAI is racing ahead in science and society, the field of medicine operates under the Hippocratic oath, whose conservative strictures require cautious, measured advancement. Our intended service to the community at this point is to blend preliminary optimism with realism. Understanding pitfalls may be more valuable than claims of transformative breakthroughs. The latter are destined to come in time, and that progress will be hastened with every pitfall that is identified and addressed.

## Figures and Tables

**Figure 1 diagnostics-15-02409-f001:**
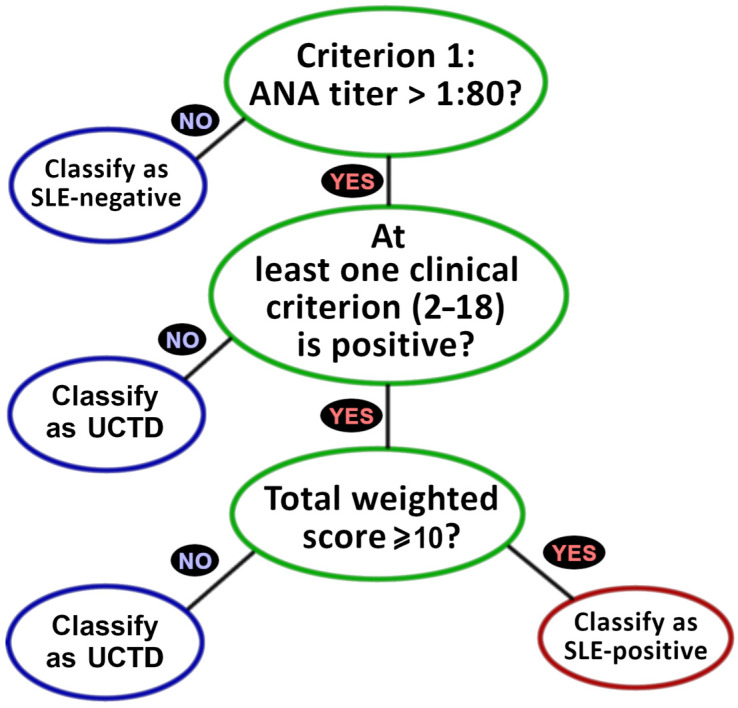
EULAR 2019 classification decision branches, beginning with the entry criterion (at least one positive recorded ANA (also known as ANA+) test required for SLE+ classification), followed by stipulation that at least one clinical criterion (Table 1, criteria 2–18) be met, and followed by the final requirement that the sum of criterial weights (the point listings in Table 1) must be 10 or greater.

**Figure 2 diagnostics-15-02409-f002:**
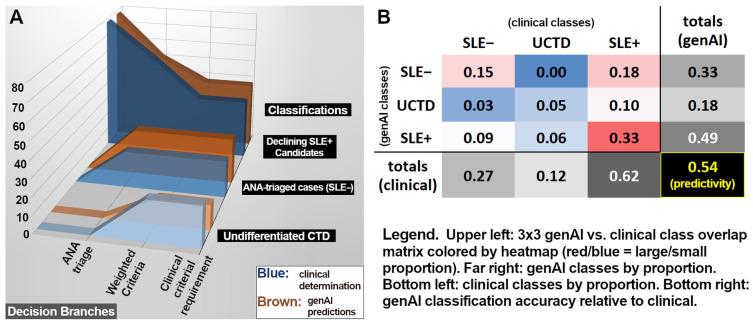
EULAR clinical and genAI classifications showing (**A**) attrition plot as a function of decision branches and (**B**) relative partitioning statistics among the classes.

**Table 1 diagnostics-15-02409-t001:** EULAR 2019 SLE criteria [20], their role in classification, and number of clinical determinations (clin-dets). Failure to meet criterion 1 confers automatic SLE− negative. SLE+ classification requires at least one positive result among clinical criteria 2–18, plus a total weighted score of at least 10.

Criteria (Abbreviation)	Role (Clin-Dets)	Data Type	Weight
1. Antinuclear antibodies (ANA)	Required (78)	quantitative test	-
2. Fever	Clinical (15)	quantitative test	2
3. Leukopenia	Clinical (15)	quantitative test	3
4. Thrombocytopenia (Thromb.)	Clinical (15)	quantitative test	4
5. Autoimmune hemolysis (AIH)	Clinical (15)	quantitative test	4
6. Delirium	Clinical (15)	qualitative exam	2
7. Psychosis	Clinical (15)	qualitative exam	3
8. Seizure	Clinical (15)	qualitative exam	2
9. Non-scarring alopecia (NSA)	Clinical (15)	qualitative exam	2
10. Oral ulcers	Clinical (15)	qualitative exam	2
11. Subacute cutan./discoid lupus (SCD)	Clinical (15)	qualitative exam	4
12. Acute cutaneous lupus (ACL)	Clinical (15)	quantitative test	6
13. Pleural/pericardial effusion (PPE)	Clinical (15)	qualitative imaging	5
14. Acute pericarditis	Clinical (15)	qualitative or imaging	6
15. Joint involvement	Clinical (15)	qualitative exam	6
16. Proteinuria	Clinical (15)	quantitative test	4
17. Lupus nephritis class II/V (LN25)	Clinical (15)	separate classification	8
18. Lupus nephritis class III/IV (LN34)	Clinical (15)	separate classification	10
19. Anti-phospholipid antibodies (APL)	Immunolog. (15)	quantitative test	2
20. Low C3 OR low C4 (C3/4)	Immunolog. (15)	quantitative test	3
21. Low C3 AND low C4 (C3+4)	Immunolog. (15)	quantitative test	4
22. Anti-dsDNA/anti-Smith antibodies (ADS)	Immunolog. (15)	quantitative test	6

**Table 2 diagnostics-15-02409-t002:** genAI accuracy for selected EULAR 2019 criteria. Criteria are defined in Table 1. PPV = positive predictive value. NPV = negative predictive value. Sensitivity and PPV are unquantifiable for criteria lacking in clinical positives and are marked n/a. **^#^** Includes 14 post-SC cases lacking explicit ANA records (ANA+ inferred from clinical SLE+ status). **^@^** Includes 4 pre-SC cases lacking explicit ANA records; ANA− inferred from clinical SLE− (non-UCTD) classification. Details available in Appendix A.

	Clinical Determination			genAI Predictions			
Key Criteria	Pos.	Neg.	Unspecified	Sens.	Spec.	PPV	NPV
1. ANA	57 **^#^**	21 **^@^**	0	0.75	0.57	0.83	0.46
3. Leukopenia	1	14	63	0	0.93	0	0.93
9. NSA	4	11	63	0.5	0.91	0.67	0.83
10. Oral ulcers	5	10	63	0.2	0.8	0.33	0.67
15. Joint involvement	2	13	63	1	0.15	0.15	1
16. Proteinuria	0	15	63	n/a	0.53	n/a	1
20. Low C3/4	0	15	63	n/a	1	n/a	1
22. ADS	0	15	63	n/a	0.87	n/a	1
Total (all 22 criteria) (± 95% c.i)	70	323	1323	0.69 ±0.11	0.87 ±0.04	0.54 ±0.10	0.93 ±0.03

**Table 3 diagnostics-15-02409-t003:** Distribution and originating evidence for ANA determinations. (TP = true positive, FN = false negative, FP = false positive, and TN = true negative).

	Total	Numerical (e.g., Titer > 1:80 or Titer < 1:80)	Phrase (e.g., “ANA Positive” or “ANA Negative”)	Not Reported in MR
TP	43	32	11	0
FN	14	0	0	14
FP	9	6	3	0
TN	12	1	7	4

**Table 4 diagnostics-15-02409-t004:** Weighted assessed influences of non-ANA EULAR-2019 criteria on SLE and UCTD determination, as defined by Equations (1a) and (1b). Additional information is available in Appendix B.

	Criterial genAI Positives		Influence [Wilson 95% Confi-dence Interval]	
	SLE+	UCTD	SLE+ (Equation (1a))	UCTD (Equation (1b))
3. Leukopenia	19	1	1.295 [0.801, 1.767]	0.300 [0.015, 1.368]
9. NSA	10	5	0.455 [0.240, 0.764]	1.000 [0.402, 1.598]
10. Oral ulcers	19	4	0.864 [0.534, 1.178]	0.800 [0.274, 1.452]
15. Joint involvement	40	8	5.455 [4.644, 5.826]	4.800 [2.652, 5.790]
16. Proteinuria	9	5	0.818 [0.412, 1.432]	2.000 [0.804, 3.196]
20. Low C3/4	14	0	0.955 [0.573, 1.431]	0.000 [0.000, 0.000]
22. ADS	23	0	3.136 [2.214, 4.038]	0.000 [0.000, 0.000]

## Data Availability

The original contributions presented in this study are included in the article. Further inquiries can be directed to the corresponding author.

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
