# Peer review of "Criteria and Protocol: Assessing Generative AI Efficacy in Perceiving EULAR 2019 Lupus Classification"

_diagnostics, 2025, doi:10.3390/diagnostics15182409_

Round 1
Reviewer 1 Report
Comments and Suggestions for Authors
The manuscript "Criteria and Protocol: Assessing Generative AI Efficacy in Perceiving EULAR 2019 Lupus Classification", by Gerald H. Lushington and colleagues, aims to investigate whether generative AI (genAI) can effectively extract and classify information from legacy-format medical records to support accurate systemic lupus erythematosus (SLE) classification, specifically by emulating the 2019 EULAR/ACR criteria. The study explores the challenges and opportunities of applying genAI to the deconvolution of heterogeneous medical data, assesses the capacity of genAI to replicate established diagnostic frameworks in terms of sensitivity and specificity, and considers the potential of such tools to alleviate information overload in clinical rheumatology by assisting in physician-guided evidence sorting. While the references are current and appropriate, and the figures and tables are well constructed, the methodology would benefit from further clarification regarding the sample size. Additionally, the absence of information on patient treatments constitutes a significant limitation, as therapeutic interventions may alter the clinical or analytical expression of classification criteria, thereby influencing AI performance. The conclusions are generally consistent with the presented evidence; however, the manuscript would be strengthened by a more explicit discussion of study limitations and potential strategies to mitigate them.
Author Response
Please refer to comments made to Referee #1 in the attached file. Thank you!

Reviewer 2 Report
Comments and Suggestions for Authors
- The affiliation and role of authors is not clear. So their expertise on the clinical topic is not clear.
- The inclusion and exclusion criteria of the study participants are nor clearly listed.
- "The present study was a retrospective analysis, ethics approval was not required for this study." This statement is concerning, since it seems that the authors think that a retrospective analysis does not need an ethical approval. This is not consistent with the international ethical standards.” A specific ethical statement is needed and, if the authors accessed the primary data collection, IRB approabl details and informed consent procedures should be described, too.
- The discussion is not deep and detailed enough.
- There is not clear conclusion at all.
Author Response
Please refer to comments made to Referee #2 in the attached file. Thank you!

Reviewer 3 Report
Comments and Suggestions for Authors
This manuscript applies a two‑pass genAI pipeline using Claude 3.5 to extract data from medical records and classify each case as SLE, UCTD, or neither, according to the 22‑criterion EULAR 2019 model. The approach is timely, but the paper would benefit from clearer emphasis on its novel contributions, methodological transparency, and careful proofreading.
1. Many prior studies have employed similar methodologies—posing clinical questions to various LLMs and scoring their answers. What unique findings or domain-specific interpretations does your study offer?
2. In Line 134, only 15 cases contained clinical validation for the remaining 21 criteria. Would the few cases affect the study's design and validation? Please clarify it.
3. In Lines 144-145, I am curious how the prompt engineering works for solving the lacking sample SLE MRs. Please clarify it.
4. In Lines 248-250, "..... reveal that specificity and NPV are greater than specificity and PPV (p < 0.02)." It appears incorrect.
5. There are several typographical errors (e.g., Lines 22, 142, 156). Please perform a thorough spell‑check.
Author Response
Please refer to comments made to Referee #3 in the attached file. Thank you!

Reviewer 4 Report
Comments and Suggestions for Authors
I appreciated having the opportunity to review this paper about the efficacy of generative AI (specifically Claude 3.5) in analysing legacy medical records to simulate classification of systemic lupus erythematosus (SLE) using EULAR 2019 criteria, I just have some suggestions that I think can help improve the manuscript overall.
- Only 15 cases had full clinical confirmation for criteria beyond ANA. This stands has a limitations on how widely the findings can be applied. This aspect needs to be stressed more in the discussion by the authors.
- Figure 2 could use clearer labels.
- Why did some ANA results were wrongly flagged as positive? please explain better, was it some labs issue?
Author Response
Please refer to comments made to Referee #4 in the attached file. Thank you!

Round 2
Reviewer 3 Report
Comments and Suggestions for Authors
Thank you for the thorough revisions. The manuscript is clearer and more coherent.